# Oxidative Stress and Epigenetics: miRNA Involvement in Rare Autoimmune Diseases

**DOI:** 10.3390/antiox12040800

**Published:** 2023-03-25

**Authors:** José Santiago Ibáñez-Cabellos, Federico V. Pallardó, José Luis García-Giménez, Marta Seco-Cervera

**Affiliations:** 1EpiDisease S.L., Scientific Park, University of Valencia, 46026 Paterna, Spain; 2U733, Centre for Biomedical Network Research on Rare Diseases (CIBERER-ISCIII), 28029 Madrid, Spain; 3Mixed Unit for Rare Diseases INCLIVA-CIPF, INCLIVA Health Research Institute, 46010 Valencia, Spain; 4Department Physiology, Faculty of Medicine and Dentistry, University of Valencia, 46010 Valencia, Spain; 5Hospital Dr. Peset, Fundación para la Investigación Sanitaria y Biomédica de la Comunitat Valenciana, FISABIO, 46010 Valencia, Spain

**Keywords:** autoimmune diseases, epigenetics, miRNAs, oxidative stress, Kawasaki disease, Sjögren’s syndrome, systemic sclerosis

## Abstract

Autoimmune diseases (ADs) such as Sjögren’s syndrome, Kawasaki disease, and systemic sclerosis are characterized by chronic inflammation, oxidative stress, and autoantibodies, which cause joint tissue damage, vascular injury, fibrosis, and debilitation. Epigenetics participate in immune cell proliferation and differentiation, which regulates the development and function of the immune system, and ultimately interacts with other tissues. Indeed, overlapping of certain clinical features between ADs indicate that numerous immunologic-related mechanisms may directly participate in the onset and progression of these diseases. Despite the increasing number of studies that have attempted to elucidate the relationship between miRNAs and oxidative stress, autoimmune disorders and oxidative stress, and inflammation and miRNAs, an overall picture of the complex regulation of these three actors in the pathogenesis of ADs has yet to be formed. This review aims to shed light from a critical perspective on the key AD-related mechanisms by explaining the intricate regulatory ROS/miRNA/inflammation axis and the phenotypic features of these rare autoimmune diseases. The inflamma-miRs miR-155 and miR-146, and the redox-sensitive miR miR-223 have relevant roles in the inflammatory response and antioxidant system regulation of these diseases. ADs are characterized by clinical heterogeneity, which impedes early diagnosis and effective personalized treatment. Redox-sensitive miRNAs and inflamma-miRs can help improve personalized medicine in these complex and heterogeneous diseases.

## 1. Introduction

Rare autoimmune diseases (rare ADs) include a broad and clinically heterogeneous group of diseases characterized by altered inflammatory and immune responses affecting any organ or system [1]. The role of epigenetics both as an important regulatory mechanism and as a potential tool for both diagnostic and therapeutic purposes is nowadays undisputed. Among all epigenetic mechanisms (and being aware of the importance of DNA methylation and histone posttranslational modifications), we have focused our attention on the role of miRNAs and their regulatory role in the inflammatory and anti-oxidant axis. We searched in the PUBMED database for systemic rare ADs with a high number of publications with the term “miRNA”. Among them, we selected Sjögren’s syndrome (SS), Kawasaki disease (KD), and systemic sclerosis (SSc), since they are the most prevalent and have the highest number of publications related to miRNAs. The selected diseases are characterized by systemic autoimmune inflammatory manifestations with persistent joint inflammation and tissue damage due to the presence of autoantibodies. Another common feature observed in ADs is oxidative stress and immune response imbalance (Figure 1). The novelty of our work is based upon choosing rare ADs with strong support of the involvement of specific key miRNAs which are controlling the expression of key transcription factors and master regulators—that control the inflammatory and antioxidant response—and then reinforcing the potential of these miRNAs as biomarkers of disease progression.

Sjögren’s syndrome (SS; MIM: 270150 and ORPHA: 289390) is a rare chronic and systemic AD recognized as autoimmune exocrinopathy and autoimmune epithelitis with female predominance (9/1 female/male predisposition). This disorder is characterized by lymphocytic infiltration consisting of activated T and B lymphocytes affecting exocrine glands (predominantly salivary and lacrimal) and autoantibody production. These factors contribute to dysfunctional secretory activity and systemic manifestations, such as rash, dry eyes and mouth, and profound fatigue, arthralgias/arthritis, nephritis, primary liver cirrhosis, and kidney and lung involvement, as well as increased risk of lymphoma [2,3]. SS may occur alone as primary SS [4] or as secondary SS in association with other ADs such as systemic lupus erythematosus (SLE) or rheumatoid arthritis [5]. Primary SS is characterized by hypergammaglobulinemia and the presence of anti-Sjögren’s syndrome antigen A (SSA)/Ro and antigen B (SSB)/La antibodies in sera. Due to its complex pathogenesis, in which genetic, epigenetic, hormonal, and environmental factors are thought to interact, no specific treatment for this disease has yet been approved [6]. Importantly, there is no effective treatment for SS: current treatments aim to alleviate disease symptoms and decrease local and systemic inflammation.

Kawasaki Disease (KD; MIM: 611775; ORPHA: 2331) is a rare acute vasculitis that produces an alteration in the walls of small- and medium-sized vessels [7]. KD occurs in children under the age of 5, and is characterized by fever, diffuse mucosal inflammation, conjunctivitis, hand and feet edema, skin rashes, and lymphadenopathy. About 20–30% of KD patients do not meet all diagnostic criteria and are considered incomplete KD. Furthermore, there is no gold standard laboratory test to help clinicians diagnose KD [8]. This is important given the severity of complications such as coronary artery lesions, coronary artery aneurysm, and myocardial infarction [9], which are all typically treated with a combination of high-dose intravenous immunoglobulin (IVIg) and oral aspirin [10]. Untreated KD patients may develop severe coronary artery aneurysms (incidence, 20–25%). Diagnosis of KD within the first 10 days of fever onset followed by treatment with IVIg can reduce coronary artery lesions [10,11], thus highlighting the critical importance of quick diagnosis for successful treatment. Despite the beneficial effects of treatment in most KD patients, about 15–25% of these patients are IVIg-resistant and have an increased risk of developing the abovementioned complications [12], which in turn complicates clinical management of KD patients. Genetic, epigenetic, and environmental factors in KD contribute to phenotypic heterogeneity and differential treatment responses. However, so far, no biomarker has been found that can effectively identify susceptibility to the disease and predict prognosis and treatment response of KD.

Systemic sclerosis (Scleroderma, SSc; MIM: 181750; ORPHA: 90291) is a chronic multisystemic disorder that causes early vascular injury, immune activation, and excessive fibrosis in the skin and internal organs [13]. This disease affects mainly women: the 7:1 female preponderance of SSc suggests that hormonal or pregnancy-related factors may have a role in SSc pathogenesis [14]. Organ disfunction due to increased tissular fibrosis is the principal cause of morbidity and mortality in this disorder [15]. Severe complications in SSc include cardiac events, pulmonary arterial hypertension, inflammatory arthritis, digital ulcers, and scleroderma renal crisis [16]. Quick diagnosis and routine evaluation of organ involvement has improved overall survival, although available treatments are directed at decreasing symptoms or avoiding organ damage and dysfunction [17]. Some genetic, epigenetic, and environmental factors have been described as contributors to the pathogenesis of SSc, but the complexity and knowledge gaps in the etiology of this disorder hinder the identification of an effective treatment addressing the molecular cause of the disease [18,19].

Given the complex etiology of ADs, epigenetics can open new avenues for identification of new molecular pathways underlying the pathologic events in these diseases, which may help identify potential therapeutic targets. Furthermore, oxidative stress, which has been linked with epigenetic changes, is a common contributor to these diseases. Against this background, in this review we outline the most recent findings concerning microRNAs (miRNAs) in SS, KD and SSc patients, highlighting the relationship with inflammation and reactive oxygen species (ROS).

## 2. Oxidative Stress in Rare Autoimmune Disorders

Increased oxidative stress in association with an inflammatory response is a common feature of the abovementioned ADs. Reactive oxygen species (ROS) and reactive nitrogen species (RNS) are produced during the normal function of cellular metabolism, acting as messengers and regulating the function of a wide array of enzymes and proteins, including those involved in the immune response. Briefly, danger signals are detected by innate immune cells, which initiate ROS and RNS production with the objective of eliminating the cause of the injury, and promoting adaptive immune system activity. Adaptative immune cells ensure that the specific cause of injury is eliminated, and then they restore homeostasis, repairing damaged cells and inhibiting pro-inflammatory signals. However, inappropriate ROS and RNS production, or a defective immune response, as occurs in autoimmune inflammation, causes permanent tissue damage due to the chemical reactivity of certain reactive compounds with biomolecules, such as DNA, lipids, carbohydrates, and proteins. Although this occurrence is widely accepted, few studies have researched the role of ROS, RNS, and antioxidant defenses in SS, KD, and SSc.

A handful of studies in SS have detected several oxidized and nitrotyrosylated biomolecules. For instance, increased levels of 8-hydroxydeoxyguanosine (8-OHdG), an oxidized nucleoside of DNA, malondialdehyde (MDA), hexanoyl-lysine (HEL, early phase oxidative stress marker) and 4-hydroxy-2-nonenal (4HNE, late phase oxidative stress marker) protein adducts, and carbonylated and nitrotyrosylated proteins were described in SS [20,21,22,23]. Despite this reported increased oxidative stress in AD, few articles have described the role of antioxidant systems in SS. There are reduced levels of total glutathione (GSH) superoxide dismutase (SOD), catalase (CAT) and glutathione peroxidase (GPX) in SS [24,25,26]. Interestingly, treatment with M3 subtype muscarinic acetylcholine receptor (M3 mAChR) autoantibodies from pSS patients on the rat submandibular acini preparation increased the specific activity of SOD and CAT [27]. Other antioxidant enzymes such as thioredoxin (TRX), peroxiredoxin 3 (PRDX3), and glutathione-S-transferase (GST) showed increased protein levels in SS [22,28,29]. Finally, some studies have evaluated the levels of enzymes related to the production of reactive species in the physiological mechanisms of the immune system. Elevated levels of enzymes that produce ROS and RNS, such as xanthine oxidase (XO), xanthine oxidoreductase (XOR), myeloperoxidase (MPO), NADPH oxidase 4 (NOX4), nitric oxide synthase 2 (NOS2), and nitric oxide synthase 3 (NOS3) were described in SS compared to healthy subjects [24,25,30,31].

There is less information in the literature about Kawasaki disease (KD) and oxidative stress than is found in SS. Straface et al. noted both an increased formation rate of nitroxyl 3-carboxy-PROXYL (CP·) and increased levels of 3-nitrotyrosine and MPO in blood from KD patients. Additionally, they described decreased levels of asymmetric dimethyl-arginine (ADMA), the endogenous inhibitor of NO synthase. These results indicated the presence of ROS and RNS in the blood of KD patients [32]. Other studies have described the presence of oxidative stress markers in serum or blood from KD patients. Specifically, Cheung et al. showed significantly higher levels of MDA and hydroperoxides in the serum of KD patients with coronary aneurysms those of controls [33]. In line with the above-mentioned study of Straface et al., hydroperoxides were observed elevated in the serum [34] and blood [35] of KS patients compared to healthy donors. Regarding antioxidant systems, one study analyzed levels of Cu-Zn SOD, Mn SOD, CAT, and GPX in blood from KD patients just after the diagnosis was established (acute phase), 2 weeks (early convalescent phase) and 4 weeks (convalescent phase) after the first blood sampling. They found decreased levels of Mn SOD, GPX, and CAT in the early convalescent phase compared to the acute phase [36]. Finally, anti-peroxiredoxin 2 antibodies have been reported in some patients with KD and the presence of these antibodies correlated with a longer duration of fever and poor response to therapy [37].

Extensive clinical and experimental evidence demonstrating increased oxidative stress has been found in SSc studies [38,39,40,41]. For instance, increased levels of several oxidative stress biomarkers such as MDA, NO, 8-isoprostane, nitrated proteins, hydroperoxides (ROOH), ADMA, and advanced oxidation protein products (AOPP) have been found in circulating blood [42,43,44,45,46,47,48], and 8-OHdG and isoprostanes have been found in urine from SSc patients [49,50,51,52]. In addition, decreased levels of total antioxidant activity were reported by Ogawa et al. [53]. Taken together, these results show a pro-oxidant status of serum/plasma and urine samples of SSc. Furthermore, in skin and cellular samples from SSc patients, increased ROS levels [48,54,55,56,57] and nitrotyrosine staining [45] were also reported. Likewise, monocytes and fibroblasts have shown elevated levels of O_2_ and H_2_O_2_ [58,59]. Finally, increased MDA and NO levels have been found in erythrocytes from SSc patients [60]. Many studies in SSc have been performed to identify the source of oxidative stress, focused mainly on NADPH oxidases (NOX, specifically NOX4) [58,61], which have been implicated in the pathogenesis of the fibrotic processes in the SSc [41,62,63,64,65]. Another finding in SSc patients is altered antioxidant systems: total antioxidant capacity [47,53], CAT, vitamin C and E, and GSH [43,47,66] showed decreased levels in plasma from SSc patients compared to controls. Interestingly, results are controversial regarding SOD status in plasma and erythrocytes from SSc patients, although more evidence has been reported suggesting impaired SOD activity [43,47,66,67]. This decrease in antioxidant capacity could be partially explained by the presence of autoantibodies in SSc patient plasma, which directly affects ROS-detoxifying enzymes such as Prdx1 [68], Cu/Zn SOD [69], and methionine sulfoxide reductase [70].

## 3. The Close Interconnexion between miRNAs and Oxidative Stress

The relationship between oxidative stress and miRNAs has attracted increasing interest in recent years. Oxidative stress-responsive miRNAs (e.g., miR-34a-5p, miR-1915-3p, miR-638, and miR-150-3p) have been found after treatment of various cell types with H_2_O_2_ [71]. In addition, cellular mechanisms regulating oxidative stress are regulated by specific miRNAs [72]; specifically, miRNAs can directly target and regulate the function of enzymes responsible for ROS and RNS production and removal. Interestingly, ROS can also affect miRNA levels, therefore showing an intricate mutual regulatory function [73,74].

Furthermore, the development, differentiation, and function of immune cells and innate and adaptive responses are partly controlled by oxidative stress [75], as well as by dynamic epigenetic mechanisms potentially modulating immune responses, which occurs also in ADs [76]. In addition, chronic inflammation produces oxidative stress [77], while vice versa, oxidative stress could induce inflammation; this may be a predisposing factor for ADs, and could in turn lead to alterations in the epigenetic machinery.

It is noteworthy that oxidative stress and inflammation can modulate both the immune response and the function of the epigenetic machinery [78]. A close relationship has also been observed between miRNAs and inflammation. Evidence provided by numerous studies indicates that miRNA expression is regulated by inflammatory stimuli and confirms the existence of feedback loops between a number of different miRNAs and several inflammatory components, such as pro-inflammatory cytokines [79,80,81,82,83], implying a close interrelationship between ROS and one of the main effectors of post-transcriptional control, miRNAs.

This ROS/miRNA/inflammation axis forms a complex trinity, contributing not only to cellular homeostasis, but also to alterations in the cellular process that can drive the organism to disease (Figure 2). Among other biomolecules taking part in this intricate axis, the specific miRNA family known as inflamma-miRs have emerged as key mediators regulating the inflammatory process, some of which have been already described as oxidative stress-responsive miRNAs, as is the case of miR-34a-5p, miR-150-3p [71], miR-155 [84], and miR-223 [85]. In a cell chondrocyte model of osteoarthritis, Cheleschi et al. demonstrated that H_2_O_2_ significantly upregulated the expression levels of SOD2, CAT, GPX, and Nrf2, also modulating miR-146a and miR-34a gene expression [86]. Interestingly, a close relationship between the inflamma-miR miR-146a-5p and oxidative stress, and this miRNA and inflammation has been described in a rat model with type 2 diabetes mellitus [87]. Moreover, depletion of the inflamma-miR miR-155 has been shown to attenuate inflammation and oxidative stress in a toll-like receptor 4 (TLR4)-dependent manner in an experimental autoimmune prostatitis model [84]. miR-21 is also reported to be regulated by oxidative stress [88].

Finally, different antioxidant therapies were also capable of modifying the expression of inflamma-miR and oxidative stress-associated miRNAs. Treatment with resveratrol [89], quercetin [90], and vitamin D [91,92] showed decreased levels of miR-155 followed by inhibition of the NF-KB signaling pathway and consequent anti-inflammatory effects. In line with this, quercetin treatment upregulated miR-146a, promoting the downregulation of inflammation via NF-KB [93]. Curcumin, an anti-proliferative, anti-inflammatory, anti-microbial, and antioxidant compound, has been described as an inductor of the oxidative stress-associated miRNA, miR-34a in different cancer models [94,95,96]. Since some antioxidant therapies could modulate both inflamma-miR and oxidative stress-associated miRNAs expression, and this is associated with decreased levels of inflammation and oxidative stress levels, it seems interesting to unravel the intricate mechanisms regulating this ROS/miRNA/inflammation axis to develop new drug therapies in rare AD.

## 4. MiRNA Alteration in Rare Autoimmune Diseases

### 4.1. Sjögren’s Syndrome

Few studies have been conducted on SS regarding the role of miRNAs in disease onset and progression. Nonetheless, some preliminary studies have provided relevant information about immune cell regulation in SS (Table 1). miR-146a and miR-155 are considered two key modulators in innate and adaptive immune response control, and both are important regulators of B- and T-cell proliferation, differentiation, and function [97]. Furthermore, both were considered inflamma-miRs activated by nuclear factor kappa B (NF-κB), which controls the Toll-like receptor/IFN pathway through receptor-associated factor 6 (TRAF6), IL-1 receptor associated kinase (IRAK1), transducer and activator transcription 1 (STA1), and interferon regulatory factor 5 (IRF5) [98]. In this regard, increased levels of miR-146a were reported in PBMCs from SS patients, which has particular importance as it regulates the innate immune and inflammatory response by repressing *IRAK1* and increasing *TRAF6* gene expression, which promotes the expression of NF-κB target genes [99]. In addition, elevated miR-146a was associated with a significantly increased percentage of Th17 cells (by targeting and negatively regulating ADAM17) in primary SS compared to healthy subjects [100]. Another study detected a high expression of inflamma-miRs miR-155-5p, miR-222-3p, miR-146a-5p and miR-28-5p in CD4^+^ T-cells, and miR-222-3p in CD19^+^ B-cells, and a low expression of let-7d-3p, miR-30c-5p and miR-378a-3p in CD4^+^ T-cells, and of miR-26a-5p, miR-30b-5p and miR-19b-3p in CD19^+^ B-cells [101]. The high levels of miR-155 in particular can be explained by the activity of the FoxP3 transcription factor, which is overexpressed in T cells infiltrating SS salivary glands, and also by the release of miR-155 from salivary gland epithelial cells; experiments performed by Le Dantec et al. showed that cultured salivary epithelial cells from SS express two-fold more miR-155 than controls [102]. In this regard, inflamma-miRs miR-155 and mir-146a inhibit Th2 proliferation and promote Th1 response. Moreover, miR-155 modulates Th17 differentiation and function and Treg differentiation [103]. These results also revealed that miR-155 increases in the PBMCs and salivary glands of SS patients [102]. Furthermore, low miR-30b-5p levels promote increased gene expression of the B-cell activating factor (*BAFF*) in the CD19^+^ cells of SS patients [101]. Increased *BAFF* expression has been correlated with autoantibody production in SS patients [104].

miR-181a is overexpressed in cultured salivary glands from SS patients [105]. In this context, Wang et al. reported increased levels of miR-181 and miR-16a in the labial salivary glands and PBMCs, which in turn was associated with a degree of inflammation in SS patients [106]. This suggests that both miRNAs are promising biomarker candidates for monitoring inflammation in SS. Moreover, miR-181a-5p levels in PBMCs of SS were associated with antigen sensitivity and exocrine gland dysfunction [107]. miR-181a-5p has been described as targeting the muscarinic receptor 3 gene (CHRM3) [108], whose genetic variants have been associated with SS [109]. Other miR-181a-5p targets include *TRIM21* and *SSB* genes, which encode for SSA/Ro and SSB/La cellular ribonucleoprotein complexes, respectively. Increased expression of miR-181a, miR-200b, and miR-223 in SS may be associated with control of elevated Ro (SSA) and La (SSB) expression [110].

A recent study performed in primary human conjunctival epithelial cells (PECs) obtained from primary SS patients identified significant differences in miR-744-5p expression at the ocular surface between pSS patients and healthy controls, and that this miRNA targets mRNA of Pellino3 (PELI3), a factor participating in regulating innate immune response through its interaction with RAK1, TRAF6, and transforming growth factor-β activated kinase 1 (TAK1). Importantly, the authors demonstrated the potential use of antago-miR-774 to restore Pellino3 expression and reduce the inflammatory response on the ocular surface mediated by IFN-dependent chemokines CCL5 and CXCL10 [111].

**Table 1 antioxidants-12-00800-t001:** miRNAs in rare autoimmune diseases.

miRNA	Rare AD	Expression	Reference
miR-155-5p	SS KD SSc	Upregulated Downregulated Upregulated	[101,102] [112] [113,114,115]
miR-146a	SS	Upregulated	[99,100,101]
miR-30b-5p	SS	Downregulated	[101]
miR-181a	SS	Upregulated	[105,106,110]
miR-200b	SS	Upregulated	[110]
miR-223	SS KD	Upregulated Upregulated	[110] [116,117,118]
miR-744-5p	SS	Upregulated	[111]
miR-200c-3p	KD	Upregulated	[119]
miR-371-5p	KD	Upregulated	[119]
miR-197-3p	KD	Upregulated	[120]
miR-125a-5p	KD	Upregulated	[121]
miR-186	KD	Upregulated	[122]
miR-27b	KD	Upregulated	[123]
miR-483	KD	Upregulated	[124]
miR-93	KD	Upregulated	[125]
miR-182	KD SSc	Upregulated Upregulated	[125] [114]
miR-296	KD	Upregulated	[125]
miR-145-5p	KD	Upregulated	[125]
miR-145-3p	KD	Upregulated	[125]
miR-27a	KD	Upregulated	[126]
miR-29a	SSc	Downregulated	[114]
miR-196a	SSc	Downregulated	[114]
miR-21	SSc	Upregulated	[114,127]
miR-150	SSc	Downregulated	[128,129]
miR-135b	SSc	Downregulated	[130]
miR-193b	SSc	Downregulated	[131]

### 4.2. Kawasaki Disease

A growing number of studies showing altered miRNA profiles in KD (Table 1) have been published in recent years, several of which have described increased levels of miR-223-3p in serum/plasma samples from KD patients [116,117,118]. Chu et al. reported that increased levels of miR-223-3p were produced and secreted by bone-marrow-derived blood cells (leukocytes and platelets), and may alter physiological conditions of vascular endothelial cells by entering cells and targeting the insulin-like growth factor 1 receptor (IGF1R) [118]. Another study in platelets of KD patients made the remarkable finding that miR-223-3p could protect against coronary artery aneurysm complications, regulating vascular smooth muscle cell (VSMC) dedifferentiation by directly targeting the platelet-derived growth factor receptor β (PDGFRβ) [132]. Furthermore, miR-197-3p levels were found to be higher in acute KD samples than in healthy controls or convalescent KD. It has been shown that miR-197-3p targets IGF1R and BCL2 transcripts in human coronary arterial endothelial cells, which modulates cell proliferation, apoptosis, and migration [120]. Increased miR-125a-5p levels in the plasma of KD patients compared to healthy control were described by Li et al., who also reported that MKK7 levels were modulated by this miRNA, thus regulating apoptosis in HUVEC cells [121]. Wu et al. described increased levels of miR-186 in serum from KD patients. Interestingly, in vitro experiments demonstrate that serum from KD patients increased the levels of miR-186 in HUVEC cells, promoting apoptosis through MAPK activation by targeting SMAD6 [122]. Similarly, miR-27b presented upregulated expression levels in KD serum and HUVEC cells treated with this serum. HUVEC-treated cells showed supressed proliferation and migration, mediated by alterations in the TGF pathway due to SMAD7 targeting by miR-27b [123]. He et al. described increased miR-483 levels in HUVEC cells treated with sera from KD patients. Moreover, the KLF4 (Krüppel-like factor 4)-miR-483-CTGF (connective tissue growth factor) axis proved useful for evaluating the pro-inflammatory and pro-endothelial-to-mesenchymal transition (EndoMT) status in KD patients. EndoMT is relevant in KD for its involvement in coronary arterial wall damage mediated by myofibroblast-like cells in these patients [124].

Other studies aimed at elucidating the role of miRNAs have been performed in immune cells. In this regard, miRNA profiles from PBMCs isolated from acute KD patients and controls showed differentially expressed miR-93, miR-182, miR-296, miR-145-5p, and miR-145-3p levels. Moreover, increased vascular endothelial growth factor A (VEGFA) mRNA in PBMCs and VEGF-A plasma levels were negatively correlated with miR-93 expression levels in the febrile phase of KD patients [125]. Luo et al. described increased miR-27a levels in B cells from patients with KD that may promote monocyte-mediated inflammatory TNF-α release by negatively regulating expression of cytoplasmic IL-10 within B-cells in vitro [126]. Other studies performed in Treg cells from KD patients showed that decreased levels of forkhead box protein 3 (FoxP3^+^) Treg might be related to downregulated miR-155 expression, leading to altered SOCS1/STAT-5 signaling and elevated miR-31 levels in patients with acute KD [112].

### 4.3. Systemic Sclerosis

The role of miRNAs in alterations to epigenetic programs has been studied in the three main systems affected in SSc patients: immune cells, endothelial cells, and fibroblasts (from both lung and skin; Table 1) [133].

A recent meta-analysis revealed the eight differentially expressed miRNAs most consistently identified across different studies performed in biospecimens obtained from SSc and healthy subjects. In dermal fibroblasts, miR-155 and miR-182 were upregulated, whereas miR-29a, miR-196a, and let7a were downregulated in SSc compared to healthy controls. miR-21 was overexpressed in serum samples [114], skin samples, and fibroblasts from SSc patients, and was associated with the expression regulation of genes involved in fibrosis, such as *SMAD7*, *SAMD3* and *COL1A1* [127]. Conversely, miR-29a was involved in inducing apoptosis in fibroblasts from SSc by regulating the Bax/Bcl-2 ratio [134] and was considered an anti-fibrotic modulator in SSc. Moreover, downregulation of miR-29a in SSc fibroblasts induces collagen deposition and its upregulation decreased the levels of type I and type III collagen [135]. Its anti-fibrotic role is also mediated through a mechanism in which miR-29a targets TGF-β-activated kinase 1 binding protein 1 (TAB1), which in turn regulates the downstream production of the tissue inhibitor of metalloproteinase (TIMP-1) [136]. These miRNAs have been implicated in immune activation and inflammation, progressive vasculopathy, and fibroblast activation and collagen production, all of which are related to fibrosis and excessive deposition of extracellular matrix (ECM) proteins found in SSc physiopathology.

Likewise, low miR-150 expression induced the expression of integrin beta3 (ITGB3), phosphorylated protein Smad3, and type I collagen, so this miRNA is proposed as an antifibrotic miRNA in SSc [128]. Moreover, downregulation of miR-150 in SSc fibroblasts upregulated ITGB3, which subsequently resulted in TGFβ pathway activation [128]. In this regard, miR-150 was also present in low levels in exosomes isolated from SSc fibroblasts [129]. Interestingly, miR-150 is among the miRNAs identified in several studies performed in SSc in both serum and dermal fibroblasts (see the meta-analysis performed by Zhang et al. [114] for more information). As occurred in SS, the inflamma-miR miR-155 was upregulated in SSc skin and lung tissue [113,115], which activated the Wnt/B-catenin and Akt signalling pathway [113], thus contributing to fibrosis. Importantly, miR-155 upregulation in SSc fibroblasts depends on NLRP3 inflammasome activation, which is required for collagen production [115]. miR-135b is reduced in fibroblasts and serum from SSc patients. Bioinformatic analysis predicts STAT6 as a target of miR-135b. This is of special relevance in autoimmunity because STAT6 is involved in the polarization of naïve T cells to Th2 effector cells, and the activation of STAT6 leads to the expression of Th2 cytokines such as IL-4 and IL-13 [130].

Finally, miR-193b has been proposed as contributor of proliferative vasculopathy in SSc, because of its role regulating urokinase-type plasminogen activator (uPA), which was demonstrated by Iwamoto et al. in fibroblasts and skin biopsies from SSc patients [131]. In this regard, miR-193b was found to be downregulated in fibroblasts and skin biopsies in SSc, which may contribute to uPA upregulation and therefore facilitate progressive vasculopathy in SSc [131].

## 5. Oxidative Stress and miRNA Dysregulation in Rare Autoimmune Disorders

The regulation of cell survival, cell growth, proliferation and differentiation and immune response by ROS and miRNAs have been extensively reported [137,138,139]. As mentioned above, both ROS and miRNAs are common features of rare autoimmune disorders. However, the crosstalk between dysregulated miRNA and the oxidative stress imbalance in these diseases is still not completely understood. This section describes the possible implications of these dysregulated miRNAs in ROS alterations.

Nuclear factor erythroid 2-related factor 2 (Nrf2), NAD-dependent protein deacetylase sirtuin-1 (SIRT1), and NF-κB signaling pathways are important mediators, regulating inflammation and oxidative stress (Figure 2). The intricate regulation of these pathways has been partially described [140,141,142], yet many players of this network are still unknown. In this scenario, miRNAs dysregulated in rare Ads can help unravel some clues.

Unsurprisingly, among all the miRNAs dysregulated in SS, KD, and SSc, miR-155, a well-known inflamma-miR, showed altered levels in all three (Table 1). Like other inflamma-miRs such as mir-146a and miR-193, this miRNA was positively regulated by NF-κB [143]. After an inflammatory stimulus, NF-κB activation rapidly promotes miR-155 expression. miR-155 then acts as an amplifier and positive regulator, repressing the expression of SH-2 containing both inositol 5′ polyphosphatase 1 (SHIP1) and the suppressor of cytokine signaling 1 (SOCS1) to ensure robust and strong NF-κB activity. Slow promotion of miR-146a levels ultimately produces a negative regulation of NF-κB activity, targeting the inducers of IRAK1 and TRAF6, which results in the attenuation of miR-155 expression and resolution of the inflammatory response [144]. Interestingly, decreased ROS production was observed when miR-155 was silenced in an endothelial cell line [145]. In contrast, overexpression of miR-155 decreased SOD and CAT activity, levels of GSH, Heme oxygenase 1 (HMOX1), and nuclear Nrf2, and increased MDA levels and cytoplasmic Nrf2 in the nucleus of mice neurons [146]. These effects could be observed in cellular models as a result of direct targeting of Nrf2 [146], SIRT1 [147], and forkhead box protein O1 (FOXO1) [148] by miR-155.

As mentioned above, increased levels of miR-223 have been reported in both KD and SS (Table 1). This miRNA has been linked to oxidative stress regulation in different studies. In a neuronal model treated with H_2_O_2_, miR-223 overexpression reduced both MDA and ROS levels, and increased SOD activity, whereas the opposite effect was observed when miR-223 was downregulated. According to the authors, this could be explained by the effect of miR-223 targeting FOXO3a and the resulting inhibition of thioredoxin interacting protein (TXNIP), whose expression is enhanced by FOXO3a [85]. TXNIP can bind and inactivate both Thioredoxin 1 (TRX1) and 2 (TRX2), which play a central role in protecting cells from oxidative stress [149]. In agreement with this protective role, Ding et al. and Zhang et al. reported that overexpression of this miRNA resulted in increased levels of Nrf2 by targeting kelch-like ECH-associated protein 1 (Keap1), the principal Nrf2 inhibitor [150,151].

Another miRNA with increased levels in KD and SSc is miR-182 (Table 1). This miRNA may play a role in oxidative stress as a protector or inducer of this imbalance. On the one hand, miR-182 has been described as a negative regulator of NOX4 [152,153], thus reducing ROS generated by this oxidase. On the other hand, it has been reported as an inhibitor of sestrin-2 (SESN2) [154,155]. Sestrin-2 activates the Nrf2 pathway by upregulating p62 and increased the interaction of Nrf2 with Keap1 [156].

Various other miRNAs with differential expression in these rare diseases have been described as regulators of Nfr2 pathway genes. Two miRNAs that are differentially expressed in KD were reported as regulators of Nfr2 expression: miR-93 [157,158] and mir-27a [159]. In addition, miR-93 targets TXNIP [158,160]. These miRNAs may therefore play a role in oxidative response regulation in KD pathogenesis. As already stated, Keap1 inhibits Nrf2 and regulates its activity, thus suggesting that other miRNAs are implicated in its regulation by targeting Keap1 or certain other proteins that modulate its inhibition. Increased levels of miR-31 were also reported in KD. Interestingly, this miRNA has been described as a regulator of fibronectin type III domain-containing 5 (FNDC5/Irisin) [161]. The hormone Irisin, which facilitates glucose uptake by skeletal muscles, improves hepatic glucose and lipid metabolism and promotes p62 expression, which in turn competes with Keap1 for binding Nrf2, increasing Nrf2 presence in the nucleus [162,163,164,165], and promotes antioxidant enzyme transcription. Another miRNA regulating Keap1 inhibition is miR-135b-5p, whose expression is decreased in SSc. This miRNA was reported to target USP15, a deubiquitinase that promotes Keap1 inhibition of Nrf2, decreasing its antioxidant activity [166,167]. In addition to upstream regulators of Nrf2, antioxidant response element (ARE) genes regulated by this protein are also targeted by miRNAs dysregulated in these rare ADs. Nrf2 regulates the expression of different components of the glutathione antioxidant system; for instance, the *SLC7A11* gene encodes the xCT light chain subunit of system xc−, which is the principal supply of cysteine necessary for GSH generation, through an antiporter mechanism that exports glutamate while importing cystine to be converted into cysteine for GSH synthesis [168]. Notably, miR-27a-3p, which was upregulated in KD, targets SLC7A11. Elevated levels of both miRNAs have been implicated in the oxidative-mediated cell death process called ferroptosis [169,170]. In the context of glutathione maintenance, both glutathione peroxidase 1 (GPX1) and 4 (GPX4) were regulated by Selenocystein (SECIS)- binding protein 2 (SBP2), and their function as detoxification enzymes is dependent on GSH. SBP2 and GPX1 were targeted by miR-181a-5p, whose levels were increased in SS [171]. Another gene regulated by Nrf2 is HMOX1, a cytoprotective enzyme involved in heme degradation, releasing iron ions, biliverdin, and CO. Bilirubin, formed via biliverdin reductase and biliverdin, is a potent antioxidant. Under normal physiological status, HMOX1 expression is suppressed by the transcription factor BTB and CNC homology 1 (Bach1). Bach1 competes with Nrf2 for binding to Maf recognition elements (MARE) in oxidative stress-response genes [172]. However, increased oxidative stress and higher heme levels inhibit Bach1-DNA binding and promote Bach1 nuclear export and degradation [173]. In line with this, one miRNA with decreased expressions in SS have been reported as regulators of Bach1: miR-30c-5p [174]. Taking all this data together, miRNA alterations could indicate an oxidative stress scenario in which the Nrf2 pathway is essential to understanding the physiopathology of these rare ADs.

Another interesting pathway related to oxidative stress is the SIRT1 pathway. SIRT1 is a NAD-dependent class III histone deacetylase that regulates the activity of many transcription factors by deacetylation [175]. SIRT1 has also been identified as a target of miR-29a [176,177], which showed decreased levels in SSc. Furthermore, it has been widely demonstrated that SIRT1 can deacetylate FoxO factors (FoxO1, FoxO3a and FoxO4) and therefore regulate the expression of antioxidant enzymes CAT, TRX, and SOD2 [178,179,180]. In addition to the abovementioned miR-155 and miR-223, which regulate FOXO1 and FOXO3a, respectively, miR-27a-3p (upregulated in KD) has been proposed as a FOXO1 regulator and modulator of oxidative stress status [181,182]. SIRT1 is closely linked to the NF-kB signaling pathway, and therefore to inflammation. Indeed, there is an antagonistic relationship between SIRT1 and NF-kB. SIRT1 can block NF-kB signaling, which deacetylates the RelA/p65 component of the NF-kB complex. In turn, the inhibition of the downstream targets of SIRT1 by NF-kB suppresses SIRT1 functions [142]. In this context, the abovementioned inflamma-miR miR-146a, with increased levels reported in SS, targets and regulates NF-kB levels through inhibition of IRAK1 and TRAF6, and blocks the pro-inflammatory signal to restore basal levels [183]. Other miRNAs found increased in SS was miR-181-5p. These miRNAs target X-linked inhibitor of apoptosis (XIAP) [184], a protein that bridges transforming growth factor β (TGFβ) and bone morphogenetic protein (BMP) receptors to TAK1, a MAP kinase kinase kinase that activates inhibitory-KB kinase (IKK), hence activating the NF-κB pathway [185,186]. Consequently, miRNA-mediated regulation of these pathways involved in oxidative stress could lead to improved understanding of these rare diseases and could be also linked to immune response.

Finally, an oxidative stress- and immune-related pathway which also regulates the three signaling pathways mentioned above is the AKT pathway. Several miRNAs altered in this rare disease were also reported to regulate members of this signaling pathway. Two upstream molecules that activate the AKT pathway by promoting phosphatidylinositol 4,5-bisphosphate (PIP2) levels are insulin grow factor 1 (IGF1) and BMP-7. These molecules were targeted by miR-29a-3p [187] and miR-135b-5p [188] respectively, and showed decreased levels in SSc. The conversion of PIP2 into phosphatidylinositol-3, 4, 5-triphosphate (PIP3) by PI3 kinase (PI3K) activates AKT, while the dephosphorylation of PIP3 into PIP2 by PTEN inactivates the signaling pathway. Conversely, the inhibitory molecule PTEN was targeted by miR-200c-3p [189] and miR-21 [190]. These miRNAs showed altered levels in KD and SSc.

## 6. Conclusions

In recent years, several studies have documented the role of epigenetic mechanisms in controlling the immune response by regulating immune cell differentiation and development, and by modulating inflammatory, anti-inflammatory, and immunosuppressive responses. The etiology of these pathologies is heterogeneous and complex, combining genetic, epigenetic, environmental, and lifestyle factors. Epigenetic studies in rare ADs such as SS, KD, and SSc have been conducted to decipher the role of miRNAs involved in autoimmunity. This review has highlighted the association of many dysregulated miRNAs reported in these diseases with oxidative stress and redox regulation, a phenotypic feature observed in these three rare ADs. Among the many miRNAs mentioned in this review, the inflamma-miRs miR-155 and miR-146 are noteworthy for their particular relevance in the inflammatory response and antioxidant system regulation. In line with this, dysregulation of one or both miRNAs have been reported in many Ads, including SS, KD, and SSc. miR-155 and miR-223 have demonstrated their important role in controlling the expression of Nrf2 and FOXO transcription factors, which confer oxidative stress resistance to cells by regulating the expression of antioxidant genes, as shown in Figure 2. Moreover, both miRNAs, miR-155 and miR-146, can control the expression of pro-oxidant and pro-inflammatory genes through the transcriptional control of NFkB.

Alterations in these miRNAs indicate the important role of adequate post-transcriptional regulation of the inflammatory response and the redox status, and their chronic alteration is a signature of AD. Nevertheless, additional studies on oxidative stress markers and other regulating molecules are clearly warranted to better elucidate the specific alterations regarding redox imbalance shown in these rare ADs.

One limitation of many epigenetic studies is that they have been conducted in the bulk of cells (PBMCs or biopsies from affected tissues), a controversial strategy given their complexity, and that different subpopulations of immune cells show specific epigenetic patterns and signatures [191,192,193].

### Translational Importance

It is clear that epigenetics, and more specifically miRNAs, will play an important role in both diagnosis and therapy. Future studies should therefore focus epigenetic analysis on specific immune cell subpopulations, to reveal which particular epigenetic changes are associated with each immune subset specifically affected in ADs. Single-cell technologies, such as mass cytometry or single-cell RNA sequencing, have revealed their potential to target different immune cell sub-populations [194,195], and future epigenetic analysis will benefit from the application of these technologies, coupled with WGBS, small-RNA sequencing, RNA sequencing, and ChIP-seq analysis in the study of autoimmune rare diseases. In addition, clarifying the interaction between epigenetics, oxidative stress and inflammation could shed light on the specific pathways occurring in each rare AD and provide specific targets which can open up the possibility of new treatments. One example of potential targets for the development of new drugs and strategies to manage these diseases is the abovementioned treatment with antago-miR-774, which is intended to reduce the inflammatory response on the ocular surface [111]. In this regard, immunomodulatory drugs, which are conceived as the principal treatment against ADs, could alter both miRNAs expression [196,197,198] and redox imbalance [199,200]. Thus, a personalized combination of both kind of drugs (ROS-sensitive miRNA modulator and immunomodulators) may set the basis of a promising strategy to treat rare ADs.

However, further research is needed to decipher the crosstalk between miRNAs, ROS, and inflammation, specifically in the physiopathology of SS, KD and SSc. ROS-sensitive miRNAs altered in rare inflammatory diseases could be inherently beneficial, not only as targets for therapeutic intervention, but also as biomarkers of diagnostic, prognostic, and disease follow-up, improving the health-related outcomes associated with these diseases.

In conclusion, a more complete understanding of how epigenetic mechanisms contribute to oxidative stress and inflammation, and conversely, of the way inflammation and oxidative stress alter miRNA regulatory programs in rare ADs, will provide valuable insight into the physiopathology, diagnostic, and therapeutic role of miRNAs in rare ADs.

## Figures and Tables

**Figure 1 antioxidants-12-00800-f001:**
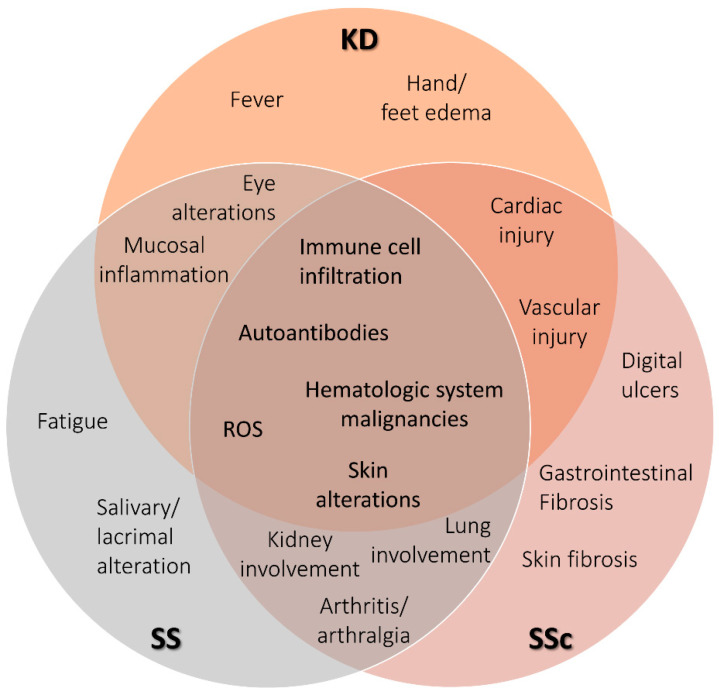
Venn diagram showing phenotypic characteristics of SS, KD, and SSc. Terms shown in the diagram are keywords defining clinical and molecular features characteristic of each rare autoimmune disease. The overlapping sections shows terms shared by the respective disease. Kawasaki disease (KD), Sjögren’s syndrome (SS), and systemic sclerosis (SSc).

**Figure 2 antioxidants-12-00800-f002:**
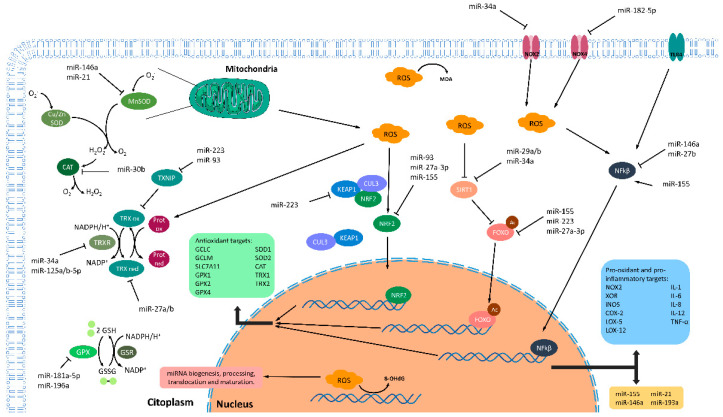
Principal interconnections of ROS, miRNAs, and inflammation. Redox imbalance is achieved by the coordinated work of ROS production and antioxidant systems. In this regard, dysregulation of these synchronized roles can produce ROS accumulation and cellular damage. In the nucleus, ROS modulate multiple steps of miRNAs metabolism including biogenesis, processing, translocation, and maturation of these small RNAs. Defective or increased ROS levels alter the normal miRNAs metabolism, consequently altering many processes regulated by them. Furthermore, increased levels of ROS affect the composition of DNA by generation of an oxidized nucleoside 8-hydroxydeoxyguanosine (8-OHdG). Nuclear localization of the transcription factors nuclear factor kappa B (NF-KB), forkhead box protein O (FOXO), and nuclear factor erythroid 2-related factor 2 (NRF2) can modulate the transcription of different antioxidant targets (green box). Furthermore, NF-KB can also modulate the pro-oxidant and pro-inflammatory targets (blue box) and the expression of some inflamma-miRNAs (yellow box). The abundance of these transcription factors in the cytoplasm, where they stay as inactive forms, is regulated by some miRNAs such as the inflamma-miR miR-155 or the family of miR-27a/b. In addition, ROS and inflammatory signals promote the nuclear translocation of these transcription factors. ROS generation is a consequence of both normal mitochondrial metabolism and inflammatory process that should be eliminated by the action of antioxidants system. In this regard, different miRNAs were described either as oxidative stress-responsive miRNAs, as is the case of miR-34a-5p, miR-150-3p, miR-155, and miR-223, or as a regulators of antioxidant enzymes (such as thioredoxin family, catalase, superoxide dismutase, glutathione peroxidases family, and glutathione metabolism-related enzymes). Induction (arrows) or inhibition (blunt-end arrows). Glutathione peroxidase (GPX); Glutathione reductase (GSR); Reduced glutathione (GSH); Oxidized glutathione (GSSG); Reactive oxygen species (ROS); Cullin-3 (CUL3); Kelch-like ECH-associated protein 1 (KEAP1); NAD-dependent protein deacetylase sirtuin-1 (SIRT1); Toll-like receptor 4 (TLR4); NADPH oxidase 2 (NOX2); NADPH oxidase 4 (NOX4); Malondialdehyde (MDA); Thioredoxin 1 (TRX1) and 2 (TRX2); Thioredoxin reductase (TRXR); Thioredoxin interacting protein (TXNIP); Catalase (CAT); Cu/Zn superoxide dismutase (Cu/Zn SOD; SOD1); Mn superoxide dismutase (Mn SOD; SOD2); Glutamate-cysteine ligase regulatory subunit (GCLM); Glutamate-cysteine ligase catalytic subunit (GCLC); Cystine/glutamate transporter (SLCA11); Xanthine oxidoreductase (XOR); Nitric oxide synthase, inducible (iNOs); Cytochrome c oxidase subunit 2 (COX-2); Polyunsaturated fatty acid 5-lipoxygenase (LOX-5); Polyunsaturated fatty acid lipoxygenase (LOX-12); Interleukin-1 (IL-1); Interleukin-6 (IL-6); Interleukin- (IL-8); Interleukin-1 (IL-12); Tumor necrosis factor alpha (TNF-α).

## Data Availability

Not applicable.

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
