# Peer review of "Oxidative Stress and Epigenetics: miRNA Involvement in Rare Autoimmune Diseases"

_antioxidants, 2023, doi:10.3390/antiox12040800_

Round 1
Reviewer 1 Report
This review has highlighted the association of abnormal expression of many miRNAs reported in three rare ADs with oxidative stress and immune disregulation.
My comments and suggestion.
Major:
1. The abstract is not informative. The roles of oxidative stress and miRNAs in rare autoimmune diseases should be briefly outlined.
2. The review includes a lot of data without their sufficient analysis and understanding. In my opinion, the MS can be shortened considerably. It would be useful to generate a table in order to understand the significance of different miRNAs in rare autoimmune diseases. It would be interesting to briefly characterize the effects of antioxidants on the expression of oxidative stress-associated miRNAs.
3. In the conclusion section, it is necessary to give a brief comparative analysis of the roles of miRNAs and oxidative stress in autoimmune and non-autoimmune (infectious) chronic inflammatory diseases. It would also be useful to describe the putative mechanisms of miRNA action on oxidative stress and vice versa.
Minor:
1) Figure 2 needs a full figure caption (description).
2) Explain what reactive oxygen metabolites (ROM) means (Line 156).
Author Response
This review has highlighted the association of abnormal expression of many miRNAs reported in three rare ADs with oxidative stress and immune disregulation.
My comments and suggestion.
Major:
- The abstract is not informative. The roles of oxidative stress and miRNAs in rare autoimmune diseases should be briefly outlined.
Following the suggestion of the reviewer the role of oxidative stress and miRNAs in rare autoimmune diseases has been outlined in the new version of the manuscript.
- The review includes a lot of data without their sufficient analysis and understanding. In my opinion, the MS can be shortened considerably. It would be useful to generate a table in order to understand the significance of different miRNAs in rare autoimmune diseases. It would be interesting to briefly characterize the effects of antioxidants on the expression of oxidative stress-associated miRNAs.
We thank the reviewer’s suggestions. We have revised the manuscript and eliminated and rewrote the paragraphs to simplify the content and make the manuscript easier to understand. Moreover, we have included a new table (table1) as suggested with the different miRNAs shared among the rare AD analyzed in the manuscript. Finally, we have included a new paragraph in “section 3” describing the close interconnexion between miRNAs and oxidative stress” indicating the effects of antioxidants on the expression of these miRNAs related to inflammation and oxidative stress.
- In the conclusion section, it is necessary to give a brief comparative analysis of the roles of miRNAs and oxidative stress in autoimmune and non-autoimmune (infectious) chronic inflammatory diseases. It would also be useful to describe the putative mechanisms of miRNA action on oxidative stress and vice versa.
Following the suggestion of the reviewer we have succinctly described in the conclusion section the impact of miR-155, miR-146 and miR-223 in the translational regulation of key factors (i.e. FOXO, NFkB and NRF2) which are controlling the anti-inflammatory and anti-oxidant response.
Minor:
- Figure 2 needs a full figure caption (description).
We thank the reviewer for his/her comment. We changed some aspects of figure 2 to improve understanding. Moreover, we have included a description in the figure legend.
- Explain what reactive oxygen metabolites (ROM) means (Line 156).
Thanks for the reviewer’s observation. In this case, authors measured reactive oxygen metabolites (ROM) with a commercial kit that essentially measured hydroperoxides (10.1186/s12872-018-0765-9; 10.1253/circj.cj-10-0605). To better understand this term, we changed the term ROM to hydroperoxides in our manuscript.
Reviewer 2 Report
This is an interesting review that evaluate the role of oxidative stress and epigenetics (via miRNA involvement) in autoimmune diseases.
I have a few comments on the review as it stands:
1. the Authors should explain why of all autoimmune diseases they have "only" choosen systemic sclerosis, Kawasaki disease and Sjogren's syndrome. There needs to be a rationale for this. It is not sufficient to say that they are rare autoimmune diseases as there are several other rare autoimmune diseases.
2. the Authors should indicate that in addition to miRNA involvement epigenetics may control autoimmune diseases in several other aspects. For example hypermethylation has been shown to suppress regulatory T cell function and a hypomethylating agent such as decitabine has been shown to improve rodent models of autoimmune diseases such as type 1 diabetes (PMID: 19841877) multiple sclerosis (PMID: 24700487), guillain Barre' syndrome (PMID: 29957387 ) and rheumatoid arthritis (PMID: 31783688)
3. Along the line of point 2, the Authors should discuss if and how immunomodulatory drugs influence the cross talk between oxidative stress, epigenetic and miRNA. Do they also envisage that this cross talk occurs for other , less rare, autoimmune diseases such as rheumatoid arthritis, SLE, type 1 diabetes, multiple sclerosis ?
4. What is the take home message fromk a translational point of view ? can these data be useful for diagnostic and therapeutic purposes ?
Author Response
This is an interesting review that evaluate the role of oxidative stress and epigenetics (via miRNA involvement) in autoimmune diseases.
I have a few comments on the review as it stands:
- the Authors should explain why of all autoimmune diseases they have "only" choosen systemic sclerosis, Kawasaki disease and Sjogren's syndrome. There needs to be a rationale for this. It is not sufficient to say that they are rare autoimmune diseases as there are several other rare autoimmune diseases.
We thank the reviewer’s observation; reviewer is absolutely right. We performed a study to select which rare autoimmune disease were more interesting to perform this manuscript. First, we checked which autoimmune diseases were also considered rare diseases. To perform this task, we used a list of autoimmune diseases from “The Global Autoimmune Institute” website (https://www.autoimmuneinstitute.org/resources/autoimmune-disease-list/) and checked if they were considered as rare disease in the principal web of rare diseases ORPHANET (https://www.orpha.net/consor/cgi-bin/Disease.php?lng=EN). With this we obtained a list of rare autoimmune diseases. Next, we checked if there were publications regarding each rare autoimmune disease and miRNAs. For this we performed a scientific search in PUBMED with the term “miRNA” and the name of each autoimmune disease from the last list (including synonyms) and we ordered by the number of manuscripts. We selected 10 with the highest number of manuscripts and performed the same search in PUBMED but with the term “oxidative stress”. With this information we selected top 5 diseases with more publications regarding miARNs excluding those with low information regarding oxidative stress. With these criteria we obtained the following rare autoimmune diseases: systemic lupus erythematosus, idiopathic pulmonary fibrosis, systemic sclerosis, Sjögren’s disease, and Kawasaki disease. Finally, we excluded idiopathic pulmonary fibrosis because it is an organ-specific rare autoimmune disease and also removed from the analysis systemic lupus erythematosus due to its complexity.
We include a paragraph in the introduction section of the manuscript in order to explain and clarify the decision to choose these rare autoimmune diseases.
- the Authors should indicate that in addition to miRNA involvement epigenetics may control autoimmune diseases in several other aspects. For example hypermethylation has been shown to suppress regulatory T cell function and a hypomethylating agent such as decitabine has been shown to improve rodent models of autoimmune diseases such as type 1 diabetes (PMID: 19841877)multiple sclerosis (PMID: 24700487), guillain Barre' syndrome (PMID: 29957387) and rheumatoid arthritis (PMID: 31783688)
Thank you to the reviewer for his/her comment. It is known that other epigenetic mechanisms (methylation of DNA and histone modification) are involved in other aspects of autoimmune diseases, however in this case we focused our work in the miRNAs-oxidative stress axis. In any case and following the reviewer’s suggestion we have included in the new version of the manuscript a statement explaining that other epigenetic factor also be also involved.
- Along the line of point 2, the Authors should discuss if and how immunomodulatory drugs influence the cross talk between oxidative stress, epigenetic and miRNA. Do they also envisage that this cross talk occurs for other, less rare, autoimmune diseases such as rheumatoid arthritis, SLE, type 1 diabetes, multiple sclerosis ?
We include information regarding the influence of immunomodulatory drugs, used in the 3 rare autoimmune diseases, in oxidative stress and miRNAs in conclusion section. This information is based in studies performed in other AD with more prevalence, such as psoriasis and rheumatoid arthritis which indicates that this cross talk occurs also in other AD. Regarding the second question, as answered in the previous questions raised by the reviewer, we have focused on systemic sclerosis, Sjögren’s disease, and Kawasaki disease according to the rationale described. Our aim was to find and describe relevant transcriptional regulation axis between miRNAs and transcriptional factors controlling the anti-oxidant and inflammatory responses. As deduced by the manuscript, this is a complex picture of the events that can occur in these rare autoimmune diseases, so we consider to include other diseases can add to much information to the manuscript and complicate the comprehension of the review.
- What is the take home message fromk a translational point of view ? can these data be useful for diagnostic and therapeutic purposes ?
We thank the reviewer for addressing this important point. Following the suggestions of the reviewer we have add in the new version of the manuscript a short paragraph entitled: “Translational importance” were we underscore the potential importance of miRNAs for diagnostic and therapeutic purposes.
Reviewer 3 Report
In this review, the authors aimed to shed light from a critical perspective on the key AD-related mechanisms, by explaining the intricate regulatory ROS/miRNA/inflammation axis and the phenotypic features of 3 autoimmune diseases. The authors state that ADs are characterized by clinical heterogeneity, which impedes early diagnosis and effective personalized treatment. In conclusion, the authors speculate that Redox-sensitive miRNAs and inflamma-miRs can help improve personalized medicine in these complex and heterogeneous diseases.
The paper is well-written and easy to follow. The contents are well-expressed, Figure 1 is crystal but figure 2 is not clear and should be redrawn.
My comments regard the rationale of the work.
As far as I know, there are dozens of autoimmune diseases characterized by chronic inflammation, oxidative stress, and autoantibodies, which cause joint tissue damage, vascular injury, fibrosis, and debilitation. Why did the authors choose Sjögren’s syndrome, Kawasaki disease, and systemic sclerosis only?
The authors should also try to underline some aspects of the novelty of the work since an expert in the field may find a few similar reviews in the scientific literature.
Author Response
In this review, the authors aimed to shed light from a critical perspective on the key AD-related mechanisms, by explaining the intricate regulatory ROS/miRNA/inflammation axis and the phenotypic features of 3 autoimmune diseases. The authors state that ADs are characterized by clinical heterogeneity, which impedes early diagnosis and effective personalized treatment. In conclusion, the authors speculate that Redox-sensitive miRNAs and inflamma-miRs can help improve personalized medicine in these complex and heterogeneous diseases.
The paper is well-written and easy to follow. The contents are well-expressed, Figure 1 is crystal but figure 2 is not clear and should be redrawn.
Thank you to the reviewer for his/her comment. We redrew some aspects of figure 2 to improve its understanding, we have included a description in the figure legend to make it clearer and easier to follow up.
My comments regard the rationale of the work.
As far as I know, there are dozens of autoimmune diseases characterized by chronic inflammation, oxidative stress, and autoantibodies, which cause joint tissue damage, vascular injury, fibrosis, and debilitation. Why did the authors choose Sjögren’s syndrome, Kawasaki disease, and systemic sclerosis only?
We want to thank the reviewer for his/her comment. In the introduction section we included a paragraph in order to make easier to understand the reason of this selection. Moreover, we want to explain more extensively the procedure by which we selected these rare autoimmune diseases.
Among the list of autoimmune diseases from “The Global Autoimmune Institute” website (https://www.autoimmuneinstitute.org/resources/autoimmune-disease-list/) we filter those considered also as a rare disease in the principal web of rare diseases ORPHANET (https://www.orpha.net/consor/cgi-bin/Disease_Search_Simple.php?lng=EN ) to obtain a rare autoimmune diseases list. Next, we performed a scientific search in PUBMEB with the term “miRNA” and the name of each autoimmune disease from the last list (including synonym) and we ordered by the number of manuscripts. With the top 10 diseases with the highest number of manuscripts we did the same search in PUBMED but with the term “oxidative stress”. Taking together all this information we selected 5 diseases with the highest number of publications in miARN and excluding those with low information regarding oxidative stress. Finally, of this list we excluded idiopathic pulmonary fibrosis because is organ-specific rare autoimmune disease and the others were systemic and systemic lupus erythematosus due to their complexity. With these criteria we selected: systemic sclerosis, Sjögren’s disease, and Kawasaki disease.
The authors should also try to underline some aspects of the novelty of the work since an expert in the field may find a few similar reviews in the scientific literature.
We thank the reviewer for his/her suggestion. We have included a statement highlighting the novelty of our review since, as far as we are aware, is the only review dealing with oxidative stress, miRNAs and rare autoimmune diseases.
Round 2
Reviewer 1 Report
The authors have significantly improved the MS. I have no more comments or suggestions
Reviewer 3 Report
The paper is quite improved. However, picture 2 is still weak. It should be improved.